# Pharmacogenetics of Donepezil and Memantine in Healthy Subjects

**DOI:** 10.3390/jpm12050788

**Published:** 2022-05-13

**Authors:** María C. Ovejero-Benito, Dolores Ochoa, Teresa Enrique-Benedito, Miriam del Peso-Casado, Pablo Zubiaur, Marcos Navares, Manuel Román, Francisco Abad-Santos

**Affiliations:** 1Clinical Pharmacology Department, Instituto de Investigación Sanitaria La Princesa (IP), Hospital Universitario de La Princesa, 28006 Madrid, Spain; mdolores.ochoa@salud.madrid.org (D.O.); te_resa_eb@hotmail.com (T.E.-B.); miriamdpeso@gmail.com (M.d.P.-C.); pablo.zubiaur@salud.madrid.org (P.Z.); marcos.navares@salud.madrid.org (M.N.); manuel.roman@salud.madrid.org (M.R.); 2Faculty of Medicine, Universidad Autónoma de Madrid (UAM), 28006 Madrid, Spain; 3Unidad de Investigación Clínica y Ensayos Clínicos (UICEC), Instituto de Investigación Sanitaria La Princesa (IP), Hospital Universitario de La Princesa, Plataforma SCReN (Spanish Clinical Research Network), 28006 Madrid, Spain; 4Centro de Investigación Biomédica en Red de Enfermedades Hepáticas y Digestivas (CIBERehd), Instituto de Salud Carlos III, 28029 Madrid, Spain; 5Instituto Teófilo Hernando, 28029 Madrid, Spain

**Keywords:** Alzheimer, pharmacogenomics, pharmacogenetic array, pharmacokinetics, anticholinergic drugs, personalized medicine, NMDA receptor antagonist

## Abstract

Donepezil and memantine are the most common drugs used for Alzheimer’s disease. Their low effectiveness could partly be explained by genetic factors. Thus, we aim to identify Single Nucleotide Polymorphisms (SNPs) associated with pharmacokinetics, pharmacodynamics, and the safety of donepezil and memantine. For this regard, 25 volunteers enrolled in a bioequivalence clinical trial were genotyped for 67 SNPs in 21 genes with a ThermoFisher QuantStudio 12K Flex OpenArray. The statistical strategy included a univariate analysis that analyzed the association of these SNPs with pharmacokinetic parameters or the development of adverse drug reactions (ADRs) followed by a Bonferroni-corrected multivariate regression. Statistical analyses were performed with SPSS software v.21 and R commander (version v3.6.3). In the univariate analysis, fourteen and sixteen SNPs showed a significant association with memantine’s and donepezil’s pharmacokinetic parameters, respectively. Rs20417 (*PTGS2*) was associated with the development of at least one ADR. However, none of these associations reached the significance threshold in the Bonferroni-corrected multivariate analysis. In conclusion, we did not observe any significant association of the SNPs analyzed with memantine and donepezil pharmacokinetics or ADRs. Current evidence on memantine and donepezil pharmacogenetics does not justify their inclusion in pharmacogenetic guidelines.

## 1. Introduction

Alzheimer’s disease (AD) is a progressive neurodegenerative disorder that slowly impairs memory and thinking skills. Although this disease has no cure, there are different approved treatments such as rivastigmine, galantamine, donepezil, and memantine [1]. These drugs are usually prescribed as combination therapy improving outcomes (function, cognition, behaviour, and global change) [1].

Memantine is an uncompetitive N-methyl-D-aspartate (NMDA) receptor antagonist that selectively blocks the excitotoxic effects of glutamate while preserving physiological transmission for normal cellular function [2,3,4,5]. Memantine shows linear pharmacokinetics when given at normal therapeutic doses. Memantine is well absorbed after an oral dose, showing an absolute bioavailability of about 100%. Its maximum concentrations are reached in about 3–7 h (Tmax) [6,7]. At a steady-state, memantine plasma concentrations vary from 70 to 150 ng/mL, showing high inter-individual variability. The mean volume of distribution is around 10 L/kg, and about 45% of the drug is bound to plasma proteins [6,7]. Memantine is mainly metabolized to three inactive compounds: 1-nitroso-3,5-dimethyl-adamantane, N-3,5-dimethyl-gludantan, and the isomeric mixture of 4- and 6-hydroxy-memantine, which are subjected to renal excretion. In vitro studies have discarded cytochrome P450 involvement in memantine’s metabolism [6,7]. Total renal clearance in healthy volunteers was 170 mL/min. Memantine’s elimination half-life (T1/2) is 60–100 h [6,7].

Donepezil hydrochloride is a piperidine derivative acetylcholinesterase inhibitor enhancing cholinergic transmission, which relieves the symptoms of mild AD [8,9]. Donepezil is slowly absorbed via the gastrointestinal tract after oral administration [10]. The maximum plasma concentration (Cmax) of donepezil is reached at 4.1 h and the mean terminal disposition T1/2 ranges from 70 to 81.5 h [11,12]. Radioactively marked donepezil studies showed that after a 5 mg oral dose of donepezil hydrochloride, 30% remains unchanged, 11% is metabolized to 6-O-desmethyl donepezil, which is also active, and the rest is made up of donepezil-cis-N-oxide (9%), 5-O-desmethyl donepezil (7%), and the glucuronide conjugate of 5-O-desmethyl donepezil (3%) [9,13]. The main enzymes involved in donepezil metabolism are CYP2D6 and CYP3A4, although CYP3A5, CYP2C9, and CYP1A2 could also be involved in this process [12,14].

The proportion of responders to donepezil and memantine were only 20–60% [15] and 30%, respectively [16]. The lack of efficacy of these drugs may be partially explained by genetic reasons [17]. Therefore, several pharmacogenetic studies searching for biomarkers of donepezil response have been performed so far [12,14,18,19,20,21,22,23,24]. There is an annotation in donepezil’s label that informs that CYP2D6 poor metabolizers had a 31.5% slower clearance than normal metabolizers [25]. Nevertheless, few studies have focused on memantine pharmacogenetics [26]. As pharmacogenetic markers can help reduce the occurrence of adverse effects, increase efficacy, and reduce costs associated with the drug and the disease [16], further studies are needed to obtain more pharmacogenetic biomarkers for donepezil and memantine. Therefore, we performed a candidate gene pharmacogenetic study including 67 polymorphisms in 21 genes, to investigate whether they affect donepezil or memantine pharmacokinetics and safety. Our study includes new genetic variants and pharmacogenes that showed an association with donepezil and memantine in previous studies.

## 2. Materials and Methods

### 2.1. Study Population

The study population comprised 30 healthy volunteers enrolled in a bioequivalence trial carried out in the Clinical Trial Unit of Hospital Universitario de La Princesa (UECHUP), Madrid (Spain). The study was approved by the Spanish Drugs Agency (AEMPS) and was conducted under current Spanish regulations and the Revised Declaration of Helsinki (https://www.wma.net/what-we-do/medical-ethics/declaration-of-helsinki, last consulted on 1 September 2021). The approved protocol and the Informed Consent Form were reviewed by the Independent Ethics Committee on Clinical Research of the Hospital Universitario de La Princesa (UECHUP-DON-MEM/18-2; EUDRA-CT: 2018-002300-14). All subjects gave their written consent for the clinical trial, while 25 of them signed the written informed consent for the pharmacogenetics study.

The inclusion criteria were: male and female volunteers aged from 18 to 55, free from organic or psychiatric conditions. The exclusion criteria were: history of kidney and/or liver damage, drug intake 48 h before receiving the study medication, having body mass index outside the 18.5–30.0 kg/m^2^ range, history of sensitivity to any drug and positive drug screening, smoker and daily alcohol consumer, blood donation and pregnant or breastfeeding women.

### 2.2. Study Design and Procedures

The study was designed as a bioequivalence clinical trial of a formulation of donepezil/memantine 10 mg/10 mg film-coated tablets compared with the pharmaceutical formulations marketed by Pfizer S.L (Aricept^®^) and by H. Lundbeck A/S (Ebixa^®^) after a single oral dose administration to healthy volunteers under fasting conditions. The clinical trial was a randomized, open-label, single-dose, single-center study crossover design, two periods, and two sequences, separated by a 28-day washout period, with the blind determination of plasma concentrations of donepezil and memantine.

The medication was administered as a single dose by oral route with 240 mL of water. Subjects fasted from 10 h before until 5 h after drug administration. Nineteen blood samples for the assessment of pharmacokinetics were obtained between baseline and 72 h after the administration of each drug. The samples were centrifuged at 4 °C for 10 min at 1900 G. The aliquots were stored in one freezer at −20 ± 5 °C until shipment to the external analytical laboratory.

Memantine and donepezil levels were measured using a high-performance liquid chromatography–mass spectrometry (HPLC–MS/MS) validated method with a limit of quantification (LLOQ) of memantine and donepezil of 99.60 pg/mL and 100.2 pg/mL, respectively.

### 2.3. Pharmacokinetic Analysis

The pharmacokinetic data of the two donepezil/memantine products were analysed by the statistical package integrated into the WinNonLin Professional Edition program, version 8.0 (Pharsight Corporation, Mountain View, CA, USA), with a non-compartmental approach.

The Cmax and time to reach the maximum plasma concentration (Tmax) were obtained directly from raw data. The elimination rate constant (ke) was calculated through linear regression of the log-linear part of the concentration–time plot. Ct is the last measured concentration. The area under the curve (AUC) was calculated from the administration to the last measured concentration (AUCt) by linear trapezoidal integration. The total AUC from administration to infinity (AUC∞) was calculated by adding the sum of AUCt and the residual area (Ct divided by ke). Values of AUC correspond to AUC∞ unless otherwise indicated. T1/2 was calculated by dividing 0.693 by ke.

The total drug clearance adjusted for bioavailability (Cl/F) was calculated by dividing the dose by the AUC∞ and adjusting for weight. The volume of distribution adjusted for bioavailability (Vd/F) was calculated as Cl/F divided by ke. AUC and Cmax were adjusted for dose and weight (AUC/dW and Cmax/dW, divided by the dose-by-weight ratio) and logarithmically transformed.

### 2.4. Safety

The safety and tolerability of memantine/donepezil were assessed by clinical evaluation of adverse events (AEs) and other parameters such as physical examinations, vital signs, and electrocardiogram. An electrocardiogram was carried out before and 2 h after drug administration. Moreover, heart rate (HR), systolic blood pressure (SBP), and diastolic blood pressure (DBP) were measured to detect abnormalities in vascular and heart function. In addition, adverse events (AEs) were noted when spontaneously reported or after an open question to the volunteer. The algorithm of the Spanish Pharmacovigilance System was used to assess causality [27]. Only those AEs that were definite, probable, or possibly related to the treatment were considered adverse drug reactions (ADRs).

### 2.5. Genotyping

DNA was extracted from 1 mL of peripheral blood samples using an automatic DNA extractor (MagNa Pure^®^ System, Roche Applied Science, Indianapolis IN, USA). Blood and DNA samples were stored at −20 °C. The genotyping strategy comprised 67 variants located in 21 genes involved in the pharmacokinetics and pharmacodynamics of donepezil or memantine (Appendix A). A customized genotyping array was carried out in an Applied Biosystems QuantStudio 12K flex qPCR instrument with an OpenArray thermal block (ThermoFisher, Waltham, MA, USA). The genotypes were assigned using small batches (Appendix A). To determine the copy number variations (CNVs) in the CYP2D6 gene, a TaqMan Copy Number Assay (Applied Biosystems, Waltham, MA, USA, Assay ID: Hs00010001_cn), which detects a specific sequence on exon 9 was used, as described previously [28]. All the samples were run in quadruplicate in a StepOnePlusTM PCR instrument (Applied Biosystems, Waltham, MA, USA). The assay was performed using an endogenous control, TaqManRNase P Copy Number Reference Assay (Assay ID: 4403326; Applied Biosystems, Waltham, MA, USA). CopyCaller Software (Applied Biosystems, Waltham, MA, USA), which applies the comparative ΔΔCT method, was used to determine the exact number of CYP2D6 copies. Genotyping was carried out in the Clinical Pharmacology Department of Hospital Universitario de la Princesa.

### 2.6. Phenotype Inference

Genotyped alleles and haplotypes were represented with an asterisk (*). *CYP2C9* (*2, *3), CYP2C19 (*2, *3, *4, *17), *CYP2B6* (*4, *5, *6, *7 and *9), *CYP2D6* (*3, *4, *5, *6, *7, *8, *9, *10, *14, *17, *41 and the number of copies), were considered to infer the enzymatic phenotype following the Clinical Pharmacogenetics Implementation Consortium (CPIC) guidelines [29,30,31,32]. CYP1A2 phenotype was inferred from *1B, *1C, and *1F variants based on previous publications [33]. *ABCB1*, C3435T, C1236T, and G2677T/A variants were classified into a haplotype as described previously [28,34].

### 2.7. Statistical Analysis

All pharmacokinetic parameters were subjected to a normality test (Shapiro–Wilk, Appendix A). Then, pharmacokinetic parameters were log-transformed to normalize distributions. AUC and Cmax were divided by the dose/weight ratio (AUC/DW, Cmax/DW) to correct the differences in weight between sexes or races, which can cause pharmacokinetic variability before logarithmic transformations.

To avoid spurious associations, a univariate analysis was carried out by comparing means of the pharmacokinetic parameters or the incidence of ADRs according to categorical variables such as sex, genotypes, or phenotypes. The means of the pharmacokinetic parameters were compared by a *t*-test (genotypes with two categories) or an ANOVA test followed by a Bonferroni post hoc (genotypes of phenotypes with three or more categories). A chi-square analysis was performed to detect significant differences in ADRs regarding sex, race, genotypes, and phenotypes. The chi-square analysis was replaced by Fisher’s exact test in cases where the number of subjects was less than 5.

Subsequently, each pharmacokinetic parameter or ADR was subjected to a multivariate analysis using linear or logistic regression, respectively. Only SNPs or phenotypes with *p* < 0.05 in the univariate analysis were included in the multivariate linear or logistic regression in addition to the possible explanatory factors such as sex and race.

The first multivariate analysis consisted of a linear regression where the dependent variables were the pharmacokinetic parameters logarithmically transformed. Genotype or phenotype variables with 3 or more categories were transformed into dummy variables: subjects with the wildtype genotype or phenotype and the other encompass all mutation carriers, either heterozygous or homozygous.

The second was a multivariate analysis of ADRs, consisting of a binary logistic regression of the incidence of each of the significant adverse effects associated with the genotypes and phenotypes in the univariate analysis plus the independent variables of sexand race.

To avoid false positives a Bonferroni correction for multiple comparisons was applied to the multivariate analysis. Thus, the significance threshold was obtained by dividing 0.05 by the number of variables introduced in the multivariate analysis. A correction for multiple comparisons was also performed using the Benjamini–Hochberg and Holm methods.

Hardy–Weinberg equilibrium (HWE) was calculated with the SNPStats software (Catalan Institute of Oncology, Barcelona, Spain) [35]. These analyses were performed using R commander (version v3.6.3) and SPSS (version 21.0, SPSS Inc., Chicago, IL, USA).

## 3. Results

### 3.1. Demographic Characteristics

These analyses were performed using R commander (version v3.6.3) and SPSS (version 21.0, SPSS Inc., Chicago, IL, USA). Our study population comprised 25 healthy volunteers (12 females and 13 males) (Table 1). Males presented a higher weight and height than females and, thus, a lower dose/weight (DW) (Table 1). No significant differences were found regarding race (data not shown).

### 3.2. Pharmacokinetics

Women showed significantly higher donepezil Vd/F levels than men (*p* = 0.015) (Table 2). Women had larger memantine AUC and Cmax values than men (*p* = 0.005 or <0.001, respectively), but these differences disappeared after adjusting by DW (Table 2). Moreover, memantine AUC/DW was higher in Latin-American volunteers (7101.4 ± 1373.68 ng∗h∗mg/mL∗kg) than in Caucasian (6259.99 ± 324.54 ng∗h∗mg/mL∗kg, *p* = 0.036).

Genotyping was successful in all subjects (Appendix A). Genotype distribution was similar between men and women except for rs10248420 (*ABCB1*, *p* = 0.031), rs10276036 (*ABCB1*, *p* = 0.025), rs7997012 (*HTR2A*, *p* = 0.026), rs3813929 and rs518147 (*HTR2C*, *p* = 0.000 and *p* = 0.001 respectively), and ABCB1 (*p* = 0.032) and CYP2C19 (*p* = 0.030) haplotypes (Appendix A). Genotype distribution was similar between Caucasian and Latin races except for rs13306278 (*COMT*, *p* = 0.015). All genetic variants fulfilled Hardy–Weinberg equilibrium except: rs2273697 (*ABCC2*, *p* = 0.027), rs2279343 (*CYP2B6*, *p* = 0.033), rs1065852 (*CYP2D6**10, *p* = 0.044), rs3892097 (*CYP2D6**4, *p* = 0.019), rs5030655 (*CYP2D6**6, *p* < 0.001), rs1414334 (*HTR2C*, *p* < 0.001), and rs518147 (*HTR2C*, *p* = 0.014).

In the univariate analysis, the following genotypes showed significant relationships with memantine’s pharmacokinetic parameters: A/G genotype for rs10248420 (*ABCB1*) with higher Cmax; T/C genotype for rs10276036 (*ABCB1*) with lower Tmax; A/G genotype for rs7787082 (*ABCB1*) with higher AUC and Cmax; G/C genotype for rs1800544 (*ADRA2*) with higher T1/2. The presence of the T allele in rs4520 (*APOC3*) and rs6314 (*HTR2A*) was correlated with lower Tmax but with higher T1/2. In addition, the G allele in rs518147 (*HTR2C*) was linked with higher Cmax. CYP2C9 IMs had lower Vd/F and higher Cmax than NMs; UMs for CYP2D6 had lower Cl/F and T1/2 than NMs (Table 3, Appendix A).

The results of the genotypes and phenotypes that did not show a significant association with memantine’s pharmacokinetic parameters in the univariate analysis are shown in Appendix A.

The detailed results of the multivariate analysis of memantine’s pharmacokinetic parameters are shown in Appendix A. After multivariate analysis, rs7787082 (*ABCB1*) was significantly associated with AUC (unstandardized β coefficients = 0.552, *p* = 0.027, R^2^ = 0.722) and with AUCinf/dW (unstandardized β coefficients = 0.460, *p* = 0.049, R^2^ = 0.476). However, none of these associations achieved the significance level (*p* = 0.004 (*p* < 0.05 divided by 11, the number of variables introduced in the multivariate analysis)) when the Bonferroni correction was applied. Non-significant results were also obtained when the Benjamini–Hochberg and Holm methods were used for multiple corrections (Appendix A).

Regarding donepezil, the presence of a C allele in rs10280101 and rs11983225 (*ABCB1*) was associated with higher Vd/F; in rs3842 (*ABCB1*) with higher Tmax and lower Cmax/dW. A G allele in rs4728709 (*ABCB1*) was correlated with lower Tmax, higher Cmax in rs7787082 (*ABCB1*) and with lower T_1/2_ in rs5128 (*APOC3*). Volunteers with one A allele in rs4680 (*COMT*) showed higher AUC/dW and T_1/2_ but lower Cl/F. Volunteers carrying the genotype A/A for rs28399433 (*CYP2A6*) had lower AUC/dW and lower Cl/F. Moreover, C allele in rs6280 (*DRD3*) was associated with higher T_max_, lower C_max_/dW; in rs6313 (*H2TRA*) with lower Tmax and higher C_max_ and C_max_/dW, and in rs518147 (*HTR2C*) with lower AUC (Table 4). IM for CYP2C9 presented higher C_max_, and IM for CYP2D6 showed higher AUC and AUC/dW and lower Cl/F compared with NM (Table 4, Appendix A). The results of the genotypes and phenotypes that were not significantly associated with donepezil’s pharmacokinetics in the univariate analysis are presented in Appendix A.

Sixteen variables were significantly related to the pharmacokinetic variability of donepezil in the univariate analysis. A multivariate analysis showed a significant relationship of donepezil’s rs6280 (*DRD3*) with AUC (unstandardized β coefficients = −0.258, *p* = 0.030, R^2^ = 0.882), with T_max_ (unstandardized β coefficients = 0.385, *p* = 0.008, R^2^ = 0.832), and with C_max_ (unstandardized β coefficients = −0.319, *p* = 0.030, R^2^ = 0.775). Moreover, we observed a correlation between donepezil’s V_d_/F and sex (unstandardized β coefficients = −0.337, *p* = 0.036, R^2^ = 0.808). Moreover, there was a link between rs4680 (*COMT*) and AUCinf/DW (unstandardized β coefficients = 0.232, *p* = 0.028, R^2^ = 0.828), and with Cl/F (unstandardized β coefficients = −0.228, *p* = 0.031, R^2^ = 0.818) (Appendix A). Nevertheless, after applying Bonferroni correction, the level of significance in the multivariate analysis was set at *p* = 0.003 (*p* < 0.05 divided by 16, the number of variables introduced in the multivariate analysis). Thus, none of these associations reached the significance level when Bonferroni correction was applied to the multivariate analysis. Similarly, non-significant results were also obtained when the Benjamini–Hochberg and Holm methods were used for multiple corrections (Appendix A).

### 3.3. Safety

Memantine and donepezil caused no clinically relevant effect on the systolic and diastolic blood pressure and heart rate. In addition, no life-threatening adverse events were registered during the clinical trial. Only seven subjects suffered an adverse drug reaction; malaise, headache, pre-syncope, dizziness, wooziness, nausea, and diarrhea. Men were less prone to present dizziness than women (16.7% vs. 66.7%; *p* = 0.029); the remaining adverse effects showed similar incidence between sex. There were no significant differences regarding race.

Volunteers carrying the C/G genotype for rs20417 (*PTGS2*) were more likely to present ADRs than C/C (*p* = 0.047, Appendix A). Moreover, the presence of the T allele in rs6265 (*BDNF*) and G in rs518147 (*HTR2C*) were associated with diarrhea (*p* = 0.024 or *p* = 0.048, respectively) (Appendix A). None of the genotypes were associated with nausea and vomiting (Appendix A). Conversely, the wild type allele (C) in rs4802419 (*CYP2B6*) or (T) in rs1800497 (DRD2) were associated with headaches or dizziness (*p* < 0.05) (Appendix A). However, all these associations were lost in the multivariate analysis (Appendix A).

Finally, a global analysis was performed to compare the presence of at least one adverse drug reaction with the absence of them. A higher Cmax in memantine and donepezil was significantly associated with the presence of adverse drug reactions (Table 5).

## 4. Discussion

Donepezil and memantine are the most common drugs used for Alzheimer’s disease. However, the percentage of responders is 20–60% for donepezil [15] and 30% for memantine [16]. Thus, it is important to search for pharmacogenetics biomarkers that could predict their efficacy and the development of adverse drug reactions. To the best of our knowledge, this is the first study that analyzed the highest number of polymorphisms associated with memantine and donepezil pharmacokinetics patterns.

We observed that females showed higher V_d_/F levels for donepezil than males, which can be explained by the higher amount of body fat in women. The effect of sex on donepezil exposure is a controversial issue [36]. Several authors have stated that men are more sensitive to donepezil treatment [37], while other authors showed the opposite effect [38]. Furthermore, women presented higher levels of memantine AUC and C_max_ than men. However, these differences disappeared after dose-weight correction. These results are in accordance with previous studies that failed to find sex-associated differences in memantine pharmacokinetics’ parameters [39].

We observed a significant association of CYP2C9 and CYP2D6 phenotypes with pharmacokinetic parameters of memantine in the univariate analysis. However, these results were not confirmed in the multivariate analysis. These results are in concordance with previous in vitro studies that showed that memantine is not a substrate for phase I metabolism (cytochrome P-450) [40]. Moreover, we observed an association in rs7787082 (*ABCB1*) and memantine AUC that was confirmed in the multivariate analysis but did not reach the Bonferroni significance threshold. These results are in accordance with a previous study that failed to find an association between *ABCB1* and memantine [26].

Donepezil is metabolized by CYP3A4 and CYP2D6 [8]. *CYP2D6* variants have been widely associated with inter-individual variability in donepezil response in AD patients [21,22,23,41]. In fact, there is an annotation in the donepezil drug label notifying that ultra-rapid metabolizers had a 24% faster clearance and that poor metabolizers of *CYP2D6* had a 31.5% slower clearance than normal metabolizers [25]. We observed that IM and PM had a 28% lower clearance and a 63.9% higher AUC than NM and UM. The reason why we did not find a significant association with them in the multivariate analysis may be due to the limited number of subjects included in the study and the reduced number of volunteers with a UM or a PM phenotype within our dataset. We did not find an association between rs4646438, rs35599367, and rs55785340 (*CYP3A4*) and pharmacokinetics in accordance with previous publications [14,24]. Moreover, we found an association between donepezil AUC, C_max_, and T_max_, and rs6280 (*DRD3*) in the multivariate analysis, but they did not reach the significance threshold. Previous studies have found an association between this SNP and the presence of behavioral disturbances in Alzheimer’s disease, although the results are controversial [42]. Moreover, there was a link between rs4680 (COMT) and AUCinf/DW and with Cl/F that did not achieve the significance threshold. Although COMT is involved in AD pathology [43], the relationship of this enzyme with the pharmacokinetic parameters of donepezil has not been described so far.

Several studies have analyzed the association of donepezil response with different genes involved in donepezil pharmacodynamics: Choline O-Acetyltransferase (*CHAT*; rs2177370, rs3793790) [44,45], Cholinergic Receptor Nicotinic Alpha 7 Subunit (*CHRNA7*; rs6494223) [46]. The SNP rs1803274 (*BCHE*), located on the butyrylcholinesterase gene was associated with donepezil response, although the results are controversial [19,45]. Finally, the *APOE* ε4/E4 genotype, associated with the greatest AD risk, appears to be associated with the worst response to drugs [24]. The reason why these genes were not included in the present study is that we focused on genes involved in donepezil and memantine pharmacokinetics. Moreover, as the study was carried out on healthy volunteers, we could not evaluate drug response.

Memantine and donepezil are usually well tolerated [6,47]. The main ADRs associated with memantine include dizziness, somnolence, headache, and constipation [48] in accordance with the ADRs reported by the participants in our current study. Moreover, donepezil can cause diarrhea, nausea, vomiting, anorexia, and muscle cramps [47]. The first three ADRs were reported by the participants in our study. As expected, we found an association between the C_max_ and the probability to experience at least one ADR.

Carriers of a mutated allele C/G in rs20417 (*PTGS2*) were more likely to suffer at least one ADR. Genetic variants of *PTGS2* are risk factors for AD [49]. Nevertheless, the role of PTGS2 in donepezil’s or memantine’s mechanism of action is still unclear. Previous studies observed that memantine can antagonize type 3 serotonin receptors (*HTR3A*) [50] and dopamine D2 receptors (*DRD2*) [51]. In concordance with these results, we observed that rs1800497 (*DRD2*) was associated with headache and dizziness. Memantine and donepezil induce the synthesis of Brain-derived Neurotrophic Factor (*BDNF*) [47,52,53]. Surprisingly, rs6265 (*BDNF*) and rs518147 (*HTR2C*) were associated with diarrhea. Intriguingly, Rs4802419 (*CYP2B6*) was linked to headaches. Although memantine is a potent CYP2B6 inhibitor [54], this enzyme does not appear to be involved in the metabolism of donepezil or memantine (Appendix A). However, none of the associations with ADRs described in this paragraph were confirmed in the multivariate analysis.

### Study Limitations

This study has some clear limitations. First, donepezil and memantine are normally used in long-term Alzheimer and our approach was based on a single dose taken by healthy volunteers. Moreover, the sample size is small but was determined by the number of subjects included in a bioequivalence trial carried out in the Clinical Trial Unit of Hospital Universitario de La Princesa. Several of the SNPs analyzed were present at a very low frequency in the Iberian or Latin population. Nevertheless, they were included in the array design because they were clinically relevant and they are part of the pharmacogenetic clinical guidelines. Thus, extreme phenotypes such as ultra-rapid metabolizers of CYP2D6 were underrepresented in the study. Thus, reduced sample size, the small effect sizes, and the genetic heterogeneity may partially explain the lack of positive results. Therefore, these findings should be confirmed in different cohorts involving a higher number of subjects. On the other hand, the advantages of our study are highly standardized conditions and there is no interference from other drugs or tobacco.

## 5. Conclusions

We did not observe any significant association of the SNPs analyzed with memantine and donepezil pharmacokinetics or ADRs. Current evidence on memantine and donepezil pharmacogenetics does not justify their inclusion in pharmacogenetic guidelines. Further research is needed to confirm the association of the different genetic variants with the pharmacokinetic parameters to adjust the dose and reduce the adverse effects. These findings may contribute to a better understanding of the inter-individual variability in donepezil and memantine pharmacokinetics and might help to improve the clinical response and tolerability of these therapies.

## Figures and Tables

**Table 1 jpm-12-00788-t001:** Demographic characteristics of the study participants.

Demographic Feature		Total (*n* = 25)	Females (*n* = 12)	Males (*n* = 13)	*p*-Value
Age (years)		30.64 ± 7.64	33.5 ± 9.62	28.00 ± 4.02	0.087
Ethnic (%)	Latin	19 (76.0%)	8 (66.7%)	11 (84.6%)	0.378
	Caucasian	6 (24.0%)	4 (33.3%)	2 (15.4%)	
Height (cm)		167.48 ± 10.51	158.42 ± 4.17	175.85 ± 6.87	0.000 *
Weight (kg)		69.58 ± 12.87	59.86 ± 7.74	78.56 ± 9.73	0.000 *
BMI (kg/m^2^)		24.63 ± 2.65	23.85 ± 2.87	25.36 ± 2.3	0.160
DW (mg/kg)		0.15 ± 0.03	0.17 ± 0.02	0.13 ± 0.02	0.000 *

Abbreviations: BMI: body mass index; DW: dose divided by weight. * *p* < 0.05.

**Table 2 jpm-12-00788-t002:** Pharmacokinetic parameters of memantine and donepezil regarding sex.

Drug	Pharmacokinetic Parameter	Females (*n* = 12)	Males (*n* = 7)	Total (*n* = 16)	*p*-Value
Donepezil	AUC (ng∗h/mL)	734.55 ± 229.74	576.33 ± 159.74	652.28 ± 208.47	0.058
	AUC/DW (ng∗h∗mg/mL × kg)	4301.37 ± 1169.75	4444.19 ± 976.87	4375.64 ± 1053.37	0.684
	*T*_max_ (h)	2.23 ± 0.76	2.37 ± 0.66	2.30 ± 0.70	0.544
	*C*_max_ (ng/mL)	20.49 ± 5.22	17.00 ± 4.48	18.67 ± 5.07	0.087
	*C*_max_/DW (ng∗mg/mL∗kg)	120.73 ± 25.91	130.49 ± 24.72	125.81 ± 25.26	0.332
	*V*_d_/*F* (L/kg)	15.87 ± 3.19	13.17 ± 2.20	14.44 ± 3.01	0.015 *
	Cl/*F* (L/h × kg)	0.25 ± 0.06	0.24 ± 0.06	0.24 ± 0.05	0.704
	*T*_½_ (h)	46.139 ± 9.96	40.34 ± 10.51	43.112 ± 10.46	0.139
Memantine	AUC (ng × h/mL)	280.42 ± 0.28	153.52 ± 0.154	263.92 ± 0.26	0.005 *
	AUC/DW (ng × h × mg/mL × kg)	1550.07 ± 1.55	969.10 ± 0.97	1253.68 ± 1.25	0.966
	*T*_max_ (h)	5.94 ± 2.26	5.12 ± 2.11	5.51 ± 2.18	0.362
	*C*_max_ (ng/mL)	18.07 ± 3.06	13.37 ± 2.28	15.63 ± 3.55	0.000 *
	*C*_max_/DW (ng × mg/mL × kg)	107.027 ± 15.67	103.33 ± 9.39	105.10 ± 12.66	0.570
	*V*_d_/*F* (L/kg)	9.53 ± 1.22	9.59 ± 0.91	9.56 ± 1.05	0.846
	Cl/*F* (L/h × kg)	0.15 ± 0.03	0.15 ± 0.022	0.15 ± 0.03	0.936
	*T*_½_ (h)	45.56 ± 10.28	45.67 ± 8.02	45.61 ± 8.98	0.882

Data are presented as mean ± SD. * *p* < 0.05.

**Table 3 jpm-12-00788-t003:** Summary of the genotypes and phenotypes that showed a significant association with memantine’s pharmacokinetic parameters in the univariate analysis.

	Genotype/Phenotype	N	AUC (ng∗h/mL)	AUC/dW (ng∗h∗mg/mL∗kg)	T_max_ (h)	C_max_ (pg/mL)	C_max_/dW (ng∗mg/mL∗kg)	*T*_½_ (h)	Vd/F (L/kg)	Cl/F (L/h∗kg)
*ABCB1* rs10248420	G/G	1	822.74 ± 0	7314.18 ± 0	7.25 ± 0	10.79 ± 0.00 *	95.9 ± 0	48.39 ± 0	9.54 ± 0	0.14 ± 0
	A/G	12	1138.78 ± 282.22	7016.85 ± 1584.48	6.17 ± 2.15	17.1 ± 2.7	105.9 ± 15.99	45.63 ± 10.36	9.45 ± 1.27	0.15 ± 0.03
	A/A	12	926.16 ± 209.36	6747.51 ± 928.86	4.71 ± 2.07	14.56 ± 3.81	105.07 ± 9.23	45.37 ± 8.23	9.68 ± 0.87	0.15 ± 0.02
*ABCB1* rs10276036	T/T	9	1138.79 ± 306.45	7165.01 ± 1717.57	6.72 ± 2.06 *	16.43 ± 3.2	103.26 ± 15.02	47.18 ± 10.91	9.56 ± 1.21	0.15 ± 0.03
	T/C	16	959.55 ± 221.42	6750.09 ± 935.56	4.83 ± 1.98	15.18 ± 3.76	106.14 ± 11.52	44.73 ± 7.95	9.56 ± 0.99	0.15 ± 0.02
*ABCB1* rs7787082	A/A	2	781.52 ± 58.3 *	6111.17 ± 1701.31	6.13 ± 1.59	12.18 ± 1.97 *^,#1^	92.92 ± 4.21	41.78 ± 9.34	9.93 ± 0.55	0.17 ± 0.05
	A/G	11	1175 ± 265.12	7208.55 ± 1508.82	6.27 ± 2.22	17.42 ± 2.59	107.35 ± 15.92	46.58 ± 10.3	9.37 ± 1.3	0.14 ± 0.03
	G/G	12	926.16 ± 209.36	6747.51 ± 928.86	4.71 ± 2.07	14.56 ± 3.81	105.07 ± 9.23	45.37 ± 8.23	9.68 ± 0.87	0.15 ± 0.02
*ADRA2A* rs1800544	C/C	7	948.63 ± 219.69	6517.46 ± 978.66	5.39 ± 2.35	15.49 ± 4.07	105.06 ± 8.47	42.6 ± 7.95 *	9.4 ± 0.66	0.16 ± 0.02
	G/C	15	989.44 ± 249.17	6808.31 ± 1122.71	4.97 ± 1.8	15.51 ± 3.67	106.28 ± 12.86	44.61 ± 8.5	9.47 ± 1.04	0.15 ± 0.03
	G/G	3	1373.34 ± 211.15	8246.54 ± 1966.97	8.5 ± 1.32	16.57 ± 2.52	99.32 ± 22.1 ^#2^	57.68 ± 3.75	10.37 ± 1.78	0.13 ± 0.03
*APOC3* rs4520	C/C	16	1045.44 ± 246.52	7110.71 ± 1270.46	6.17 ± 2.12 *	15.06 ± 2.87	102.38 ± 11.15	47.91 ± 9.2	9.74 ± 1.04	0.15 ± 0.02
	C/T	9	986.11 ± 304.11	6523.89 ± 1200.34	4.33 ± 1.82	16.64 ± 4.54	109.95 ± 14.35	41.53 ± 7.33	9.25 ± 1.05	0.16 ± 0.03
*CYP2C9*	NM	21	1009.08 ± 280.65	6984.66 ± 1341.13	5.71 ± 2.22	14.84 ± 3.17 *	103.03 ± 11.86	46.94 ± 9.03	9.73 ± 0.98 *	0.15 ± 0.03
	IM	4	1102.82 ± 149.36	6452.13 ± 515.02	4.44 ± 1.8	19.78 ± 2.59	115.97 ± 12.5	38.64 ± 4.95	8.67 ± 1.1	0.16 ± 0.01
*CYP2D6*	UM + NM	2	976.08 ± 243.70 *	6723.09 ± 1053.94∗	5.43 ± 2.31	15.15 ± 3.56	104.24 ± 12.98	44.70 ± 8.68	9.60 ± 1.10	0.15 ± 0.02
	IM+ PM	2	1276.10 ± 246.00 *	7825.39 ± 1951.29 *	5.94 ± 1.46	18.12 ± 2.60	109.62 ± 11.21	50.44 ± 10.29	9.38 ± 0.86	0.13 ± 0.03
*HTR2A* rs6314	C/C	21	1012.16 ± 285.94	6751.67 ± 1277.64	5.61 ± 1.95	15.99 ± 3.72	106.59 ± 11.25	43.57 ± 8.12 *	9.35 ± 0.86 *	0.15 ± 0.03
	C/T	4	1086.67 ± 77.14	7675.35 ± 857.71	5 ± 3.47	13.71 ± 1.71	97.29 ± 18.45	56.33 ± 4.79	10.69 ± 1.37	0.13 ± 0.02
*HTR2C* rs518147	C/C	11	839.26 ± 152.51 *^,#3^	6550.63 ± 919.57	5.14 ± 2.22	13.38 ± 2.32 *^,#3^	103.84 ± 8.3	42.93 ± 6.49	9.47 ± 0.76	0.16 ± 0.02
	G/C	6	1165.53 ± 306.44	6843.25 ± 1843.45	6.29 ± 1.61	18.01 ± 3.11	105.38 ± 11.4	44.82 ± 10.17	9.55 ± 0.88	0.15 ± 0.04
	G/G	8	1172.12 ± 206.5 ^#4^	7421.25 ± 1117.26	5.44 ± 2.57	16.94 ± 3.73	106.62 ± 18.77	49.9 ± 10.46	9.7 ± 1.54	0.14 ± 0.02
	Total	25	1024.08 ± 263.92	6899.46 ± 1253.68	5.51 ± 2.18	15.63 ± 3.55	105.1 ± 12.65	45.61 ± 8.98	9.56 ± 1.05	0.15 ± 0.03

* *p* < 0.05 after ANOVA or *t*-test compared to the other category. ^#1^: *p* < 0.05 after ANOVA and Bonferroni post hoc (A/A vs. A/G), ^#2^: *p* < 0.05 after ANOVA and Bonferroni post hoc (C/C vs. G/G), ^#3^: *p* < 0.05 after ANOVA and Bonferroni post hoc (C/C vs. C/G), ^#4^: *p* < 0.05 after ANOVA and Bonferroni post hoc (C/C vs. G/G).

**Table 4 jpm-12-00788-t004:** Donepezil pharmacokinetic parameters based on genotypes or phenotypes with significant variability in the univariate analysis.

	Genotype/Phenotype	N	AUC (ng∗h/mL)	AUC/dW (ng∗h∗mg/mL∗kg)	Tmax (h)	C_max_ (pg/mL)	C_max_/dW (ng ×*mg/mL∗kg)	*T*_½_ (h)	V_d_/F (L/kg)	Cl/F (L/h∗kg)
*ABCB1* rs10280101	A/A	16	668.75 ± 242.41	4573.19 ± 1159.48	2.36 ± 0.7	18.44 ± 5.06	128.1 ± 24.7	42.55 ± 12.21	13.4 ± 1.91 *	0.23 ± 0.06
	A/C	9	622.99 ± 136.71	4024.44 ± 768.91	2.19 ± 0.73	19.08 ± 5.35	121.74 ± 27.22	44.13 ± 6.86	16.29 ± 3.78 *	0.26 ± 0.05
*ABCB1* rs11983225	T/T	16	668.75 ± 242.41	4573.19 ± 1159.48	2.36 ± 0.7	18.44 ± 5.06	128.1 ± 24.7	42.55 ± 12.21	13.4 ± 1.91 *	0.23 ± 0.06
	T/C	9	622.99 ± 136.71	4024.44 ± 768.91	2.19 ± 0.73	19.08 ± 5.35	121.74 ± 27.22	44.13 ± 6.86	16.29 ± 3.78 *	0.26 ± 0.05
*ABCB1* rs3842	T/T	20	670.65 ± 223.03	4446.73 ± 1117.81	2.13 ± 0.60 *	19.69 ± 4.78 *	131.48 ± 21.66 *	42.61 ± 11.24	13.97 ± 2.57	0.24 ± 0.06
	T/C	5	578.77 ± 126.59	4091.26 ± 772.05	3 ± 0.68 *	14.6 ± 4.41 *	103.15 ± 28.23 *	45.15 ± 7.04	16.31 ± 4.19	0.25 ± 0.05
*ABCB1* rs4728709	A/G	8	638.47 ± 255.84	4450.28 ± 1250.16	2.69 ± 0.58 *	16.48 ± 4.04	117.5 ± 23.82	43.32 ± 13.48	13.97 ± 1.93	0.24 ± 0.05
	G/G	17	658.77 ± 190.82	4340.51 ± 988.18	2.12 ± 0.69 *	19.7 ± 5.28	129.72 ± 25.65	43.02 ± 9.19	14.66 ± 3.43	0.24 ± 0.06
*ABCB1* rs7787082	A/A	2	485.06 ± 58.65	3810.89 ± 1230.23	2.25 ± 1.06	12.01 ± 2.85 *^,#1^	90.92 ± 2.9	46.59 ± 8.47	17.98 ± 2.67	0.28 ± 0.09
	A/G	11	710.3 ± 198.75	4354.73 ± 1164.04	2.32 ± 0.74	20.18 ± 3.97 *	124.64 ± 25.56	44.35 ± 11.08	15 ± 3.39	0.24 ± 0.05
	G/G	12	626.96 ± 222.36	4488.92 ± 989.49	2.29 ± 0.68	18.4 ± 5.52 *	132.7 ± 22.86	41.41 ± 10.66	13.33 ± 2.16	0.24 ± 0.05
*APOC3* rs4520	C/C	16	682.54 ± 217.81	4610.74 ± 1129.47	2.47 ± 0.75	17.57 ± 4.14	119.57 ± 22.68	47.08 ± 10.47 *	14.92 ± 2.77	0.23 ± 0.05
	C/T	9	598.48 ± 190.61	3957.67 ± 793.57	2 ± 0.5	20.63 ± 6.17	136.9 ± 27.08	36.08 ± 5.97 *	13.58 ± 3.39	0.26 ± 0.05
*APOC3* rs5128	C/C	20	670.73 ± 209.34	4476.06 ± 1081.81	2.33 ± 0.74	18.31 ± 4.87	122.79 ± 25.84	45.49 ± 10.11 *	14.93 ± 3.05	0.24 ± 0.05
	C/G	5	578.48 ± 209.97	3973.95 ± 919.99	2.2 ± 0.57	20.09 ± 6.17	137.88 ± 20.7	33.63 ± 5.45 *	12.48 ± 2	0.26 ± 0.06
*COMT* rs4680	G/G	12	558.64 ± 193.87 *	3786.77 ± 761.8 *	2.21 ± 0.53	18.13 ± 4.47	124.78 ± 20.6	37.03 ± 7.25 *	14.34 ± 2.81	0.27 ± 0.05 *
	G/A	13	738.71 ± 188.69 *	4919.21 ± 1010.12 *	2.38 ± 0.84	19.17 ± 5.7	126.76 ± 29.75	48.74 ± 9.96 *	14.53 ± 3.29	0.21 ± 0.04 *
*CYP2A6* rs28399433	A/A	19	615.71 ± 180.08	4077.07 ± 865.3 *^,#2^	2.26 ± 0.76	18.77 ± 5.03	124.82 ± 25.52	41.61 ± 9.25	14.93 ± 3.08	0.26 ± 0.05 *^,#2^
	A/C	4	872.46 ± 243.77	5557.56 ± 1128 *	2.38 ± 0.48	19.6 ± 6.71	123.7 ± 32.52	52.05 ± 12.33	13.54 ± 2.49	0.19 ± 0.03 *
	C/C	2	559.35 ± 210.72	4848.16 ± 1267.87 *	2.5 ± 0.71	15.86 ± 2.27	139.46 ± 3.08	39.61 ± 15.6	11.53 ± 1.56	0.22 ± 0.05 *
*CYP2C9*	NM	21	631.89 ± 214.21	4364.55 ± 1119.04	2.38 ± 0.73	17.52 ± 4.32 *	122.36 ± 25.34	43.68 ± 11.23	14.62 ± 2.99	0.24 ± 0.06
	IM	4	759.31 ± 153.58	4433.81 ± 723.12	1.88 ± 0.25	24.7 ± 4.79 *	143.94 ± 17.41	40.17 ± 4.54	13.46 ± 3.34	0.23 ± 0.03
*CYP2D6*	UM + NM	21	591.81 ± 152.02 *	4093.18 ± 820.11 *	2.31 ± 0.76	18.22 ± 5.32	125.34 ± 26.43	40.62 ± 8.90 *	14.56 ± 3.26	0.25 ± 0.05 *
	IM + PM	4	969.71 ± 183.48 *	5858.51 ± 950.00 *	2.25 ± 0.29	21.03 ± 2.72	128.28 ± 25.26	56.22 ± 8.61 *	13.80 ± 0.92	0.18 ± 0.03 *
*DRD3* rs6280	T/T	6	710.97 ± 154.29	4522.5 ± 782.39	1.67 ± 0.26 *^,#3^	23.09 ± 5.83	144.42 ± 18.5 *^,#4^	40.8 ± 10	13.03 ± 2.34	0.23 ± 0.04
	T/C	10	651.34 ± 248.12	4135.9 ± 1011.25	2.35 ± 0.66 *	17.2 ± 4.06	111.81 ± 17.16 *	43.97 ± 8.43	15.57 ± 2.15	0.26 ± 0.06
	C/C	9	614.19 ± 204.99	4544.1 ± 1294.52	2.67 ± 0.7 *	17.36 ± 4.23	128.96 ± 29.21 *	43.72 ± 13.43	14.12 ± 3.91	0.24 ± 0.06
*HTR2A* rs6313	C/C	12	665.26 ± 226.85	4376.74 ± 1232.73	2.02 ± 0.41 *^,#5^	20.1 ± 5.34 *^,#5^	132.9 ± 26.24 *^,#5^	42.17 ± 10.6	14.17 ± 2.59	0.25 ± 0.06
	C/T	9	683.34 ± 220	4544.29 ± 1003.05	2.22 ± 0.74∗	19.22 ± 4.16 *	128.59 ± 15.79 *	44.2 ± 11.01	14.09 ± 2.34	0.23 ± 0.05
	T/T	4	543.42 ± 102.69	3992.87 ± 584.04	3.31 ± 0.38 *^,#6^	13.15 ± 2.24 *	98.3 ± 26.12 *	43.52 ± 11.52	16.02 ± 5.4	0.26 ± 0.04
*HTR2C* rs518147	C/C	11	522.27 ± 145.24 *^,#7^	4047.75 ± 797.42	2.32 ± 0.55	16.57 ± 4.62	127.92 ± 25.94	38.24 ± 10.4	13.63 ± 2.5	0.26 ± 0.05
	C/G	6	779.75 ± 244.12 *	4566.77 ± 1444.99	2.21 ± 0.84	20.13 ± 6.45	116.95 ± 29.85	46.79 ± 11.48	15.23 ± 3.03	0.24 ± 0.07
	G/G	8	735.43 ± 167.05 *^,#8^	4683.12 ± 1046.98	2.34 ± 0.86	20.46 ± 3.92	129.55 ± 22.35	47.07 ± 7.72	14.96 ± 3.69	0.23 ± 0.05
	Total	25	652.28 ± 208.47	4375.64 ± 1053.37	2.3 ± 0.7	18.67 ± 5.07	125.81 ± 25.26	43.12 ± 10.46	14.44 ± 3.01	0.24 ± 0.05

Data are presented as mean ± SD. * *p* < 0.05 after ANOVA or *t*-test compared to the other category. ^#1^: *p* < 0.05 after ANOVA and Bonferroni post hoc (A/A vs. A/G), ^#2^: *p* < 0.05 after ANOVA and Bonferroni post hoc (A/A vs. A/C), ^#3^: *p* < 0.05 after ANOVA and Bonferroni post hoc (T/T vs. C/C), ^#4^: *p* < 0.05 after ANOVA and Bonferroni post hoc (T/T vs. T/C), ^#5^: *p* < 0.05 after ANOVA and Bonferroni post hoc (C/C vs. T/T), ^#6^: *p* < 0.05 after ANOVA and Bonferroni post hoc (C/T vs. T/T), ^#7^: *p* < 0.05 after ANOVA and Bonferroni post hoc (C/C vs. C/G), ^#8^: *p* < 0.05 after ANOVA and Bonferroni post hoc (C/C vs. G/G).

**Table 5 jpm-12-00788-t005:** Global analysis of adverse effects associated with pharmacokinetic parameters.

Drug	Pharmacokinetic Parameter	No AEs (*n* = 10)	AEs (*n* = 15)	Total (*n* = 25)	*p*-Value
Donepezil	AUC (ng∗h/mL)	583.12 ± 174.53	698.38 ± 221.86	652.28 ± 208.47	0.175
	AUC/DW (ng∗h∗mg/mL∗kg)	4308.93 ± 992.26	4420.11 ± 1124.23	4375.64 ± 1053.37	0.822
	*T*_max_ (h)	2.38 ± 0.78	2.25 ± 0.66	2.30 ± 0.70	0.747
	*C*_max_ (pg/mL)	16.26 ± 4.60	20.28 ± 4.84	18.67 ± 5.07	0.041 *
	*C*_max_/DW (ng∗mg/mL∗kg)	119.32 ± 20.2	130.14 ± 27.95	125.81 ± 25.26	0.382
	*V*_d_/*F* (L/kg)	14.47 ± 3.16	14.42 ± 3.01	14.44 ± 3.01	0.988
	Cl/*F* (L/h∗kg)	0.25 ± 0.06	0.24 ± 0.05	0.24 ± 0.05	0.824
	*T*_½_ (h)	42.67 ± 10.72	43.42 ± 10.65	43.12 ± 10.46	0.837
Memantine	AUC (ng∗h/mL)	901.07 ± 216.56	1106.09 ± 266.92	1024.08 ± 263.92	0.049
	AUC/DW (ng∗h∗mg/mL∗kg)	6683.05 ± 1092.15	7043.73 ± 1368.14	6899.46 ± 1253.68	0.504
	*T*_max_ (h)	5.68 ± 1.93	5.4 ± 2.38	5.51 ± 2.18	0.581
	*C*_max_ (ng/mL)	13.46 ± 2.75	17.07 ± 3.35	15.63 ± 3.55	0.007 *
	*C*_max_/DW (ng∗mg/mL∗kg)	99.39 ± 8.14	108.91 ± 13.89	105.1 ± 12.65	0.084
	*V*_d_/*F* (L/kg)	9.84 ± 0.92	9.37 ± 1.12	9.56 ± 1.05	0.254
	Cl/*F* (L/h∗kg)	0.15 ± 0.03	0.15 ± 0.03	0.15 ± 0.03	0.501
	*T*_½_ (h)	45.69 ± 9.44	45.56 ± 9	45.61 ± 8.98	0.980

* *p* < 0.05.

## Data Availability

The data that support the findings of this study are available upon reasonable request.

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
