# Peer review of "Pharmacogenetics of Donepezil and Memantine in Healthy Subjects"

_jpm, 2022, doi:10.3390/jpm12050788_

Round 1

Reviewer 1 Report

The authors present a pharmacogenetic study of donepezil and memantine in healthy volunteers. This topic is discussed with similar results in several articles cited by the authors.

I consider the 25 subjects to be very few to draw a meaningful conclusion, because  Chi-hong Tseng & Yongzhao Shao (2010) in their article entitled Sample Size Analysis for Pharmacogenetic Studies, Statistics in Biopharmaceutical Research, 2:3, 319-328, DOI: 10.1198/sbr.2009.08076  specifically suggest the number of patients depending on the frequency of different genotypes.

My suggestion is to increase the number of cases depending on the frequency of occurrence of the genotypes discussed.

Author Response

Thank you very much for your comments. We agree with the reviewer that the reduced sample size used in this study is one of the main limitations of this publication. Nevertheless, the current study was carried out thanks to a bioequivalence clinical trial conducted in the Clinical Trial Unit of Hospital Universitario de La Princesa (UECHUP), Madrid (Spain). Therefore, the number of subjects recruited was restricted by the number of subjects enrolled in the bioequivalence study who agreed to sign the informed consent. We are afraid that is not feasible to increase the sample size. We have modified the study limitations section to emphasize this fact.

Reviewer 2 Report

This paper describes a study on the pharmocogenetics of two Alzheimer's drugs, donepezil and memantine. The authors collected genotype and pharmocokinectic data on a volunteer cohort. An association of several SNPs with pharmacokinetic parameters of both drugs are reported. Regarding ADRs, one SNP-ADR association is reported. However, these associations are not significant after multiple testing correction.

The paper is well writen,  and the description and discussion of the results are appropriate. Conclusions are in line with the results.

The major issue of this paper is the lack of significant results.

Several factors can influence this and and I feel a more thorough discussion of this matter is warranted. Small sample size, which the authors identify as a limitation is possibly one important factor.

Regarding statistical significance I have one main question for the authors: did you try any other multiple correction methods? As you might be aware, Bonferroni is often too stringent and might yield false negatives. In this case I would sugest that other correction mehods, such Benjamini-Hochberg or Holm could be used. Do these improve the results?

Author Response

Thank you very much for your comments. We have modified the study limitation paragraph to discuss different factors that could influence the results of the pharmacogenetic study such as genetic heterogeneity or small effect sizes. Several of the genotyped SNPs have a very low frequency in the Iberian or Latin population. Nevertheless, they were included in the array design because they are used to infer the metabolizer phenotypes in the pharmacogenetic clinical guidelines. Thus, extreme phenotypes such as ultra-rapid metabolizers of CYP2D6 were underrepresented in the study. All the factors mentioned above may partially explain the lack of positive results. This information was included in the first paragraph of page 11.

According to the reviewer’ suggestion, we repeated all the multivariate analyses applying the Holm and the FDR correction (Supplementary Tables 4 and 6). Please see the attachment. However, we failed to find any significant result in any of the analyses performed. We have modified the materials and methods and the results sections to include this information. Nevertheless, we consider that the publication of negative results is very important in the pharmacogenetic area because the creation of new pharmacogenetic clinical guidelines depends on the validation of the potential biomarkers by different research groups.

Round 2

Reviewer 1 Report

I agree the corrections.

Reviewer 2 Report

I feel the authors addressed my concerns regarding the study.